# Prognostic Value of Creatinine Levels at Admission on Disease Progression and Mortality in Patients with COVID-19—An Observational Retrospective Study

**DOI:** 10.3390/pathogens12080973

**Published:** 2023-07-25

**Authors:** Antonio Russo, Mariantonietta Pisaturo, Caterina Monari, Federica Ciminelli, Paolo Maggi, Enrico Allegorico, Ivan Gentile, Vincenzo Sangiovanni, Vincenzo Esposito, Valeria Gentile, Giosuele Calabria, Raffaella Pisapia, Canio Carriero, Alfonso Masullo, Elio Manzillo, Grazia Russo, Roberto Parrella, Giuseppina Dell’Aquila, Michele Gambardella, Antonio Ponticiello, Lorenzo Onorato, Nicola Coppola

**Affiliations:** 1Infectious Diseases, Department of Mental Health and Public Medicine, University of Campania “L. Vanvitelli”, 80138 Naples, Italy; antonio.russo2@unicampania.it (A.R.); mariantonietta.pisaturo@unicampania.it (M.P.); caterina.monari@gmail.com (C.M.); federica.ciminelli1987@gmail.com (F.C.); lorenzo.onorato@unicampania.it (L.O.); 2Infectious Diseases Unit, A.O. S Anna e S Sebastiano Caserta, 81100 Caserta, Italy; paolo.maggi@unicampania.it; 3Emergency Unit, P.O. Santa Maria delle Grazie, 80078 Pozzuoli, Italy; enrico.allegorico@aslnapoli2nord.it; 4Infectious Disease Unit, University Federico II, 80138 Naples, Italy; ivan.gentile@unina.it; 5Third Infectious Diseases Unit, AORN dei Colli, P.O. Cotugno, 80131 Naples, Italy; sangio.vincenzo@gmail.com; 6IV Infectious Disease Unit, AORN dei Coli, P.O. Cotugno, 80131 Naples, Italy; vincenzoesposito@ospedalideicolli.it; 7Hepatic Infectious Disease Unit, AORN dei Colli, P.O. Cotugno, 80131 Naples, Italy; valeria.gentile25@gmail.com; 8IX Infectious Disease Unit, AORN dei Coli, P.O. Cotugno, 80131 Naples, Italy; g.calabria@tin.it; 9First Infectious Disease Unit, AORN dei Coli, P.O. Cotugno, 80131 Naples, Italy; raffipisapia@gmail.com; 10Department of Infectious Diseases, AORN S. Pio “G. Rummo” General Hospital, 82100 Benevento, Italy; canio.carriero@ao-rummo.it; 11Infectious Disease Unit, A.O. San Giovanni di Dio e Ruggi D’Aragona Salerno, 84131 Salerno, Italy; al.masullo@alice.it; 12VIII Infectious Disease Unit, AORN dei Coli, P.O. Cotugno, 80131 Naples, Italy; manzillo@libero.it; 13Infectious Disease Unit, Ospedale Maria S.S. Addolorata di Eboli, ASL Salerno, 84025 Eboli, Italy; gr.russo@aslsalerno.it; 14Respiratory Infectious Disease Unit, AORN dei Colli, P.O. Cotugno, 80131 Naples, Italy; roberto.parrella@ospedalideicolli.it; 15Infectious Disease Unit, A.O. Avellino, 83100 Avellino, Italy; dellaquilagiuseppina@libero.it; 16Infectious Disease Unit, P.O. S. Luca, Vallo della Lucania, ASL Salerno, 84078 Salerno, Italy; gambardella1960@gmail.com; 17Pneumology Unit, AORN Caserta, 81100 Caserta, Italy; antonio.ponticiello@unina.it

**Keywords:** creatinine, kidney disease, COVID-19, SARS-CoV-2 infection, outcome, mortality

## Abstract

Introduction: Acute kidney disease and chronic kidney disease are considered conditions that can increase the mortality and severity of COVID-19. However, few studies have investigated the impact of creatinine levels on COVID-19 progression in patients without a history of chronic kidney disease. The aim of the study was to assess the impact of creatinine levels at hospital admission on COVID-19 progression and mortality. Methods: We performed a multicenter, observational, retrospective study involving seventeen COVID-19 Units in the Campania region in southern Italy. All adult (≥18 years) patients, hospitalized with a diagnosis of SARS-CoV-2 infection confirmed by a positive reverse transcriptase-polymerase chain reaction on a naso-oropharyngeal swab, from 28 February 2020 to 31 May 2021, were enrolled in the CoviCamp cohort. Results: Evaluating inclusion/exclusion criteria, 1357 patients were included. Considering in-hospital mortality and creatinine value at admission, the best cut-off point to discriminate a death during hospitalization was 1.115 mg/dL. The logistic regression demonstrated that factors independently associated with mortality were age (OR 1.082, CI: 1.054–1.110), Charlson Comorbidity Index (CCI) (OR 1.341, CI: 1.178–1.526), and an abnormal creatinine value at admission, defined as equal to or above 1.12 mg/dL (OR 2.233, CI: 1.373–3.634). Discussion: In conclusion, our study is in line with previous studies confirming that the creatinine serum level can predict mortality in COVID-19 patients and defining that the best cut-off of the creatinine serum level at admission to predict mortality was 1.12 mg/dL.

## 1. Introduction

Since 2019, severe acute respiratory syndrome coronavirus-2 (SARS-CoV-2) has spread all over the world and caused a global crisis in healthcare and the economic and social sectors. [1]. The disease can cause severe illness, and it is characterized by a high mortality rate in certain groups [2,3]. Several studies have highlighted that different clinical conditions, such as diabetes, hypertension, chronic kidney disease, and dementia, can increase the mortality and severity of the disease [2,3]. Identifying high-risk patients shortly after admission allows for timely, supportive treatment [2,4,5].

Although the lungs are the most commonly affected organs by SARS-CoV-2, the kidneys are also frequently affected. In fact, acute kidney disease is a frequent complication of Coronavirus Disease-2019 (COVID-19), especially in patients with a severe disease [6,7]. The interplay between a hyperactive immune response, the cytopathic effects of the virus, and homeostatic reactions to balance the pulmonary hemodynamic response have been postulated to be the main cause of kidney damage [8]. As a predictor of poor outcome, creatinine levels at hospital admission may be a useful parameter to quickly identify high-risk COVID-19 patients who require intensive management [9,10]. To our knowledge, few studies have investigated the impact of creatinine levels on COVID-19 progression in patients without a history of chronic kidney disease [11].

Considering the study published earlier, the aim of this observational, retrospective study on a large cohort of COVID-19 patients hospitalized from February 2020 to May 2021 was to assess the impact of creatinine levels at hospital admission on COVID-19 disease progression and mortality in subjects without underlying chronic kidney disease.

## 2. Materials and Methods

### 2.1. Study Design and Setting

A multicenter, observational, retrospective study was performed, involving seventeen COVID-19 units in eight cities in the Campania region in southern Italy: Naples, Caserta, Salerno, Benevento, Avellino, Pozzuoli, Eboli, and Vallo della Lucania. Fifteen out of 17 were Infectious Disease Units, while the remaining 2 were sub-intensive care units; all these centers had collaborated in previous studies and were able to perform oxygen treatment and non-invasive ventilation (NIV).

All adult patients (≥18 years old) admitted from 28 February 2020 to 31 May 2021 with a diagnosis of SARS-CoV-2 confirmed by a positive reverse transcriptase-polymerase chain reaction (RT-PCR) on a naso-oropharyngeal swab were included in our CoviCamp cohort. From the CoviCamp cohort, we included in the present study all patients for whom a determination at admission of creatinine and severity of COVID-19 were available and who did not have chronic kidney disease (CKD). The exclusion criteria included minority age and lack of clinical data and/or of informed consent. No study protocol or guidelines regarding the criteria of hospitalization were shared among the centers involved in the study, and the patients were hospitalized following a decision by the physicians of each center.

The study was approved by the Ethics Committee of the University of Campania L. Vanvitelli, Naples (n°10877/2020). All procedures performed in this study were in accordance with the ethics standards of the institutional and/or national research committee and with the 1964 Declaration of Helsinki and its later amendments or comparable ethics standards. Informed consent was obtained from all participants included in the study.

This study was reported following the STROBE recommendations for an observational study (see Appendix A).

### 2.2. Variables and Definitions

All demographic and clinical data of patients with SARS-CoV2 infection enrolled in the cohort were collected in an electronic database. From this database we extrapolated the data for the present study.

The microbiological diagnosis of SARS-CoV-2 infection was defined as a positive RT-PCR test on a naso-oropharyngeal swab.

We divided the patients enrolled according to the clinical outcome of COVID-19 during hospitalization: mild, moderate, severe outcome, and death. Patients with mild infection did not need oxygen (O_2_) therapy and/or had a Modified Early Warning Score (MEWS) below 3 points during hospitalization. Patients with a moderate infection required non-invasive O_2_ therapy (excluding high-flow nasal cannula) and/or had a MEWS score equal to or above 3 points (≥3) during hospitalization. Patients with a severe infection needed management in an intensive care unit (ICU) and/or high-flow nasal cannula or invasive/non-invasive mechanical ventilation during hospitalization.

Patients were followed until negativity of SARS-CoV-2-RNA on naso-oropharyngeal swab and/or until they were discharged from hospital or died. The glomerular filtration rate (eGFR) was calculated using 2021 CKD-EPI [12]. Chronic Kidney Disease (CKD) was defined considering the diagnoses of CKD prior to hospitalization.

### 2.3. Statistical Analysis

For the descriptive analysis, categorical variables were presented as absolute numbers and their relative frequencies. Continuous variables were summarized as mean and standard deviation or as median and interquartile range (Q1–Q3). We performed a comparison of the patients who were discharged from hospital and those who died during hospitalization using Pearson chi-square or Fisher’s exact test for categorical variables and Student’s *t* test or Mann–Whitney tests for continuous variables. The same statistical analyses were performed to compare patients with different cut-offs of creatinine blood levels at admission. Odds ratios were calculated using binomial logistic regression; these analyses were performed only for parameters clinically relevant and for those that were statistically significant in univariate analysis. The receiver operating characteristic (ROC) curve and the Youden’s Index were used to determine the optimum cut-off point for possible effective variables on the patients’ outcomes. A *p*-value below 0.05 was considered statistically significant. Analyses were performed using STATA [13].

## 3. Results

During the study period, 2054 patients with a diagnosis of SARS-CoV-2 infection, confirmed by a positive RT-PCR on a naso-oropharyngeal swab, were hospitalized. Among them, 191 were excluded because they already suffered from CKD, and 506 were excluded for the lack of data on the creatinine value; thus, 1357 patients were included in the present study (Figure 1).

The demographic and clinical characteristics of the 1357 patients included in the study are shown in Table 1, Table 2, Table 3 and Table 4. Eight hundred and twenty-six (60.9%) subjects were males, the median age was 62 years (Q1–Q3: 51–73 years), the median creatinine value at admission was 0.83 mg/dL (Q1–Q3: 0.7–1 mg/dL), and the median eGFR was 94 mL/min (Q1–Q3: 74–105 mL/min). No patients had received vaccination for SARS-CoV-2. A total of 641 (47.6%) patients had a mild outcome, 299 (22%) a moderate outcome, 308 (22.7%) a severe outcome, and 104 (7.7%) died (Table 3).

We compared the patients discharged (n = 1253) with those who died during hospital stay (n = 104) (Table 4). The patients who died during hospitalization were significantly older (*p* = 0.0001), with a higher CCI (*p* = 0.0001), higher rate of cardiovascular diseases (50 vs. 23.7%, *p* = 0.0001) and dementia (43 vs. 23%, *p* = 0.0001), had a shorter hospitalization stay (*p* = 0.0001), a lower P/F ratio (*p* = 0.0001), a lower median of days from symptom onset to hospital admission (*p* = 0.0001), and a higher creatinine value at admission (*p* = 0.0001) (Table 4 and see Appendix A). Other biochemical and clinical data are shown in Table 4.

Considering these data, we calculated the Area Under the Curve (AUC) for creatinine level at admission considering death during hospitalization as a status variable. The result was 0.647 (95% Confidence Interval (CI): 0.582–0.713; *p* < 0.001), with the best cut-off point of 1.115 mg/dL (sensitivity: 44.2%, specificity: 85.7%) demonstrating the direct correlation between death during hospitalization and creatinine levels at admission (see Appendix A).

Considering this cut-off, we defined two groups: a case group with a creatinine level equal to or higher than 1.12 mg/dL at admission (225 patients, 16.6%) and a control group with a creatinine level lower than 1.12 mg/dL at admission (1132 patients, 83.4%). The subjects in the case group were significantly older than those in the control group (*p* ≤ 0.001), had a higher median of CCI (*p* = 0.0001), and had more comorbidities, such as arterial hypertension (*p* = 0.001), cardio-vascular disease (*p* = 0.001), diabetes (*p* = 0.001), and COPD (Chronic obstructive pulmonary disease) (*p* = 0.001) (Table 5). There were no statistical differences between the case and control groups in the prevalence of chronic liver disease (*p* = 0.826), in median days from symptom onset to hospital admission (*p* = 0.696), and in median PaO_2_/FiO_2_ Ratio (P/F) at admission (*p* = 0.074) (Table 5). Moreover, the patients in the case group exhibited a higher rate of severe clinical outcome of COVID-19 (45.3% vs. 27.4%, *p* = 0.001) and of in-hospital mortality (20.4% vs. 5.1%, *p* = 0.001) (Table 5).

Lastly, the logistic regression demonstrated that the factors independently associated with mortality were age (OR 1.082, CI: 1.054–1.110), CCI (OR 1.341, CI: 1.178–1.526), and an abnormal creatinine value at admission, defined as equal to or above 1.12 mg/dL (OR 2.233, CI: 1.373–3.634) (Table 6)

## 4. Discussion

In this multicenter observational retrospective study, the aim was to assess the impact of creatinine levels at admission on disease progression and mortality in subjects without underlying chronic kidney disease. Compared to those with normal values, the patients with abnormal serum creatinine at hospital admission exhibited a higher severe COVID-19 rate and mortality rate; moreover, in the present large cohort of hospitalized COVID-19 patients, the best cut-off of creatinine serum levels at admission to predict mortality was 1.12 mg/dL.

Several studies have investigated the role of creatinine levels or renal function on COVID-19 clinical outcome and on mortality, demonstrating a significant association between high levels of creatinine and COVID-19 severity and mortality [8,10,11,14,15,16,17,18,19,20,21,22,23,24,25,26]. A systematic review performed by Izcovich and colleagues in 2020 aimed at identifying factors associated with a more severe COVID-19 presentation, which may help clinicians in the decision-making process of patients with COVID-19 [9]. Among the 207 studies included, the authors identified a serum creatinine increase as a prognostic factor for mortality increase (OR 1.14, 95% CI 1.02–1.28), but not for disease severity (OR 1.89, 95% CI 0.87–4.10) [9].

However, the majority of the published studies did not exclude or state that they were excluding patients with CKD, which does not allow us to know whether the creatinine value predicted the COVID-19 outcome in subjects without CKD, such as in the present study. A recent retrospective cohort study performed in Italy on 174 hospitalized patients with COVID-19 investigated the role of kidney dysfunction on COVID-19 severity and mortality, and examined in-depth the relationship between kidney function, age, and progression of COVID-19. The authors reported acute kidney impairment (AKI) in 10.2% of patients, and an overall mortality rate of 19.5% [14]. Patients with AKI presented a higher risk of mortality (OR 5.16, 95% CI 1.86–14.30) compared with non-AKI patients, especially in moderate–severe AKI (stages 2–3, aHR 8.43, 95% CI 2.96–24.02) compared to mild AKI (stage 1, aHR 1.90, 95% CI 0.62–5.82) [14]. Another predictor of mortality was age ≥ 70 years (aHR 6.2, 95% CI 1.80–21.40). However, the influence of baseline eGFR on clinical outcome decreased with age, and it did not seem to represent a risk factor for patients older than 70 years [14].

Another retrospective observational study, by Alfano et al., demonstrated that COVID-19 patients with decreased kidney function within 24 h from hospital admission were at high risk for AKI and in-hospital mortality [10]. The authors included 224 non-ICU patients, evaluated 24-h serum creatinine differences (ΔsCr), and stratified them in three groups: The first had stable kidney function (ΔsCr from –0.05 mg/dL up to +0.05 mg/dL), the second had decreased kidney function (ΔsCr higher than +0.05 mg/dL), and the last had improved kidney function (ΔsCr lower than −0.05 mg/dL). ΔsCr was a predictor of AKI (HR 7.9, *p* < 0.001) and mortality during hospitalization (HR 4.0, 95%, *p* < 0.001) [10]. A higher survival rate was observed in the first group, whereas the second one was associated with higher rates of both AKI and in-hospital mortality [10]. The regression analysis demonstrated that decreased kidney function was an independent risk factor for 30-day in-hospital mortality (HR 5.5, *p* = 0.04) [10].

To our knowledge, only one study investigated the role of creatinine serum levels in COVID-19 progression focusing only on patients without a history of CKD [11]. Chen et al. investigated the effect of abnormal renal function on prognosis of 1764 COVID-19 patients, considering not only serum creatinine values, but also cystatin C values, which is a more sensitive indicator of GFR compared with BUN (Blood Urea Nitrogen) and serum creatinine [11]. The authors demonstrated that elevated cystatin C levels were associated with a more severe clinical outcome (OR 2.449, *p* < 0.001), whereas elevated serum creatinine levels were independent predictors of death (OR 6.789, *p* 0.02) [11].

The creatinine value and renal function are very useful parameters not only in the evaluation and prognosis of patients with COVID-19, but also in antiviral treatment for SARS-CoV-2 infection. To date, remdesivir is the only antiviral approved in patients hospitalized for COVID-19 [27] and is not recommended for patients with eGFR < 30 mL/min [27]. However, recent observational studies and systematic reviews [28,29,30,31,32] reevaluate the impact of remdesivir in patients with reduced renal function or eGFR < 30 mL/min, suggesting that remdesivir can also be used in these patients if the potential benefits outweigh the risks [27].

Our study has several limitations. Firstly, it has a retrospective design. Secondly, we evaluated only patients with at least one value of creatinine serum at baseline, but we did not consider the trend of creatinine, nor blood urea nitrogen and/or cystatin C. Thirdly, we did not take into consideration other potential causes of increased values of serum creatinine, such as bacterial superinfection or dehydration caused by diarrhea. Fourthly, due to the historical period of our study, our cohort of patients did not include subjects who were vaccinated against SARS-CoV-2 and/or early treatments (antivirals or monoclonal antibodies). Moreover, our study did not account for the impact of different variants with a possible impact on the applicability of the findings to the vaccinated patients, which represents the majority of patients nowadays. On the other hand, the strengths of our study are the multicenter design and the sample size of the population.

In conclusion, our study is in line with previous studies in confirming that the creatinine serum level can predict mortality in COVID-19 patients and in defining that the best cut-off of creatinine serum levels at admission to predict mortality is 1.12 mg/dL. Despite the significant data, it should be noted that the study population had greater comorbidities and CCI compared to the control group, which, even if not statistically significant, may be clinically significant in determining an increase in creatinine and consequently with mortality and severity.

Despite the recent improvements in preventive and therapeutic measures, it is of paramount importance to find other examinations that can help clinicians to foresee the clinical progression of COVID-19. Although the studies are retrospective observational studies, they all demonstrate that the baseline creatinine level and its variations, especially in the first 24 h, seem to be a predictor of poor outcome and may be a useful parameter to identify high-risk patients requiring more careful attention.

## Figures and Tables

**Figure 1 pathogens-12-00973-f001:**
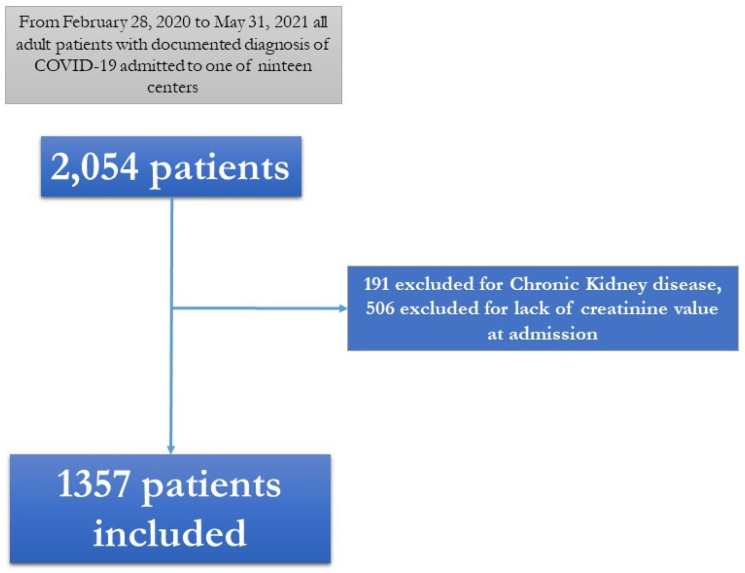
**Figure 1**. Flow chart of patients included in the study.

**Table 1 pathogens-12-00973-t001:** Demographic, clinical and laboratory data of the patients included at admission.

		N° of Patients with Data Available
Males *	826 (60.9)	1357
Age, years **	62 (51–73)	1357
Charlson comorbidity index **	2 (1–4)	1219
Days from symptom onset to admission in hospital **	7 (3–10)	789
Patients with fever in recent history *	822 (60.9)	1349
Patients with dyspnea in recent history *	923 (68.5)	1347
Patients with asthenia in recent history *	397 (29.5)	1346
Patients with cough in recent history *	459 (34.1)	1346
Patients with ageusia/dysgeusia in recent history *	53 (3.9)	1347
Patients with anosmia/hyposmia in recent history *	45 (3.3)	1347
Patients with diarrhea in recent history *	56 (4.2)	1344
Patients with skin lesions in recent history *	6 (0.4)	1336
White blood cells (WBC) at admission **	8100 (5830–10,810)	1338
International Normalized Ratio (INR) at admission **	1.1 (1.02–1.19)	1249
Blood creatinine at admission **	0.83 (0.7–1)	1357
Patients with abnormal serum creatinine value ^£,^*	443 (32.6)	1357
eGFR at admission **	94 (74–105)	1357
ALT at admission **	33 (21–54.5)	1329
AST at admission **	32 (22–47)	1236
Total bilirubin at admission **	0.6 (0.44–0.84)	1239
Creatinine phosphokinase (CPK) at admission **	82 (48–169)	925
Lacticodehydrogenase (LDH) at admission **	300 (235–410)	1284
PaO_2_/FiO_2_ Ratio (P/F) at admission **	239 (145–317)	1178

Footnotes: £ Abnormal value defined above 0.95 mg/dL; * N° (%) of patients; ** Median (Q1–Q3).

**Table 2 pathogens-12-00973-t002:** Comorbidity of patients included in our study.

		N° of Patients with Data Available
Patients with hypertension *	626 (46.1)	1357
Patients with cardiovascular disease *	349 (25.7)	1357
Patients with diabetes *	246 (18.1)	1357
Patients with chronic obstructive pulmonary disease *	119 (8.8)	1356
Patients with chronic liver disease *	45 (3.3)	1351
Patients with malignancy *	85 (6.3)	1355
Patients with dementia *	66 (4.9)	1351

Footnotes: * N° (%) of patients.

**Table 3 pathogens-12-00973-t003:** Clinical outcome of patients included in our study.

		N° of Patients with Data Available
Patients with mild outcome *	641 (47.6)	1357
Patients with moderate outcome *	299 (22.0)	1357
Patients with severe disease *	308 (22.7)	1357
Patients who died during hospitalization *	104 (7.7)	1357
Days from admission to discharge ^$,^**	14 (9–21)	1344

Footnotes: $ or death in patients who died during hospitalization; * N° (%) of patients; ** Median (Q1–Q3).

**Table 4 pathogens-12-00973-t004:** Demographic, clinical and laboratory parameters in patients discharged or who died during hospitalization.

	Patients Dead During Hospitalization N°: 104	Patients Discharged from Hospital N°: 1253	*p*-Value
Males *	56 (53.8)	770 (61.5)	0.127 ^a^
Age, years **	82 (75–86.5)	61 (50–71)	0.0001 ^b^
Charlson comorbidity index **	5 (4–6)	2 (1–4)	0.0001 ^c^
Days from symptom onset to admission in hospital **	3 (0–7)	7 (4–10)	0.0001 ^c^
Patients with hypertension *	61 (58.7)	565 (45.1)	0.008 ^a^
Patients with cardiovascular disease *	52 (50)	297 (23.7)	0.0001 ^a^
Patients with diabetes *	30 (28.8)	216 (17.2)	0.003 ^a^
Patients with chronic obstructive pulmonary disease *	16 (15.4)	103 (8.2)	0.013 ^a^
Patients with chronic liver disease *	6 (5.8)	39 (3.1)	0.149 ^a^
Patients with malignancy *	14 (13.5)	71 (5.7)	0.002 ^a^
Patients with dementia *	23 (22.1)	43 (3.4)	0.0001 ^a^
Patients with obesity *	12 (17.6)	102 (10.2)	0.054 ^a^
PaO_2_/FiO_2_ Ratio (P/F) at admission **	134 (100–242)	248 (153–320)	0.0001 ^c^
Days from admission to discharge ^$,^**	10 (6.5–15.5)	15 (10–21)	0.0001 ^c^
Blood creatinine at admission **	1 (0.78–1.45)	0.8 (0.7–1)	0.0001 ^c^
eGFR at admission **	68.6 (41.5–88.7)	95.3 (76.7–105.5)	0.0001 ^b^

Footnotes: $ or death in patients who died during hospitalization; * N° (%) of patients; ** Median (Q1–Q3); a, Chi-square test; b, Student’s *t*-test; c, Mann Whitney test.

**Table 5 pathogens-12-00973-t005:** Demographic, clinical, and laboratory parameters in patients with creatinine at admission <1.12 and ≥1.12 mg/dL.

	Patients with Creatinine Value More than or Equal to 1.12 mg/dL (Case Group) N°: 225	Patients with Creatinine Value Less than 1.12 mg/dL (Control Group) N°: 1132	Univariate Analysis
			*p*-Value
Males *	164 (72.9)	662 (58.5)	0.001 ^a^
Age, years **	73 (64–82)	59 (50–71)	0.001 ^b^
Charlson comorbidity index **	4 (3–5)	2 (1–4)	0.001 ^c^
Days from symptom onset to hospital admission **	7 (2–10)	7 (3–10)	0.696 ^c^
Patients with hypertension *	142 (63.1)	484 (42.8)	0.001 ^a^
Patients with cardio-vascular disease *	106 (47.1)	243 (21.5)	0.001 ^a^
Patients with diabetes *	70 (31.1)	176 (15.5)	0.001 ^a^
Patients with chronic obstructive pulmonary disease *	33 (14.7)	86 (7.6)	0.001 ^a^
Patients with chronic liver disease *	8 (3.6)	37 (3.3)	0.826 ^a^
Patients with malignancy *	17 (7.6)	68 (6)	0.374 ^a^
Patients with dementia *	15 (6.7)	51 (4.5)	0.169 ^a^
Patients with obesity *	18 (10.2)	96 (10.8)	0.812 ^a^
PaO_2_/FiO_2_ Ratio (P/F) at admission **	220 (142–305)	245 (145–320)	0.074 ^c^
Days from admission to discharge ^$,^**	14 (9–21)	14 (10–21)	0.472^c^
Patients with severe clinical outcome *	102 (45.3)	310 (27.4)	0.001 ^a^
Patients who died during hospitalization *	46 (20.4)	58 (5.1)	0.001 ^a^

Footnotes: $ or death in patients who died during hospitalization; * N° (%) of patients; ** Median (Q1–Q3); a, Chi-square test; b, Student’s *t*-test; c, Mann–Whitney test.

**Table 6 pathogens-12-00973-t006:** Multivariable binomial logistic regression of parameters predictor of mortality.

	OR	95% Lower Confident Interval	95% Upper Confident Interval	*p*-Value
Gender	1.075	0.665	1.737	0.768
Age, years	1.082	1.054	1.110	0.001
Charlson comorbidity index	1.341	1.178	1.526	0.001
Patients with hypertension	0.780	0.486	1.254	0.305
Patients with cardiovascular disease	0.985	0.607	1.599	0.952
Patients with diabetes	0.794	0.461	1.369	0.407
Patients with chronic obstructive pulmonary disease	0.678	0.347	1.325	0.255
Patients with abnormal serum creatinine value **	2.233	1.373	3.634	0.001

Footnotes: ** Abnormal value defined as equal to or above 1.12 mg/dL.

## Data Availability

The data presented in this study are available on request from the corresponding author.

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
