# Peer review of "Prognostic Value of Creatinine Levels at Admission on Disease Progression and Mortality in Patients with COVID-19—An Observational Retrospective Study"

_pathogens, 2023, doi:10.3390/pathogens12080973_

Round 1
Reviewer 1 Report
In this observational retrospective study Russo A et al wanted to assess the impact of serum creatinine at hospital admission on COVID-19 disease progression and mortality in subjects without chronic kidney disease. The authors report that abnormal serum creatinine at hospital admission can predict in-hospital mortality, with a cut-off of 1.12 mg/dl. This is an interesting analysis that involve a large multicentric cohort of COVID-19 patients.
The paper is potentially interesting but needs some additional edits and clarifications prior to considering for publication.
1) The paper dose not respect the submission guidelines provided by the Journal, in particular: abstract should include a brief background (before stating the aim of the study) and should not include headings; references (for journal articles) should be reported as: Author 1, A.B.; Author 2, C.D. Title of the article. Abbreviated Journal Name Year, Volume, page range. Please check instructions for the authors and correct bibliography
2) Reference number 4 is not about mortality in specific groups of COVID patients but talk about prediction role of liver enzymes at hospital admission on disease progression and clinical outcome. The authors should remove it and cite a proper paper.
2) Number 17 and number 22, are duplicated reference, please correct and renumber.
3) In the “introduction” section, line 73-82 should be removed
4) the authors enrolled 1357 patients admitted to a COVID unit in Campania from March 2020 to May 2021. This long time-period embraces the first three Italian pandemic waves in which the clinical course of the disease and therapeutic approach changed significantly. The authors should possibly specify in the “study design” the setting of care (did they include also Intensive Care Units?) and at least oxygen therapy (conventional oxygen supply or NIMV?) since it is an indirect marker of disease severity.
5) the authors says that eGFR was calculated according to the latest guidelines (line 120). Anyway, the reference proposed (number 11) does not refer to guidelines. Please, specify which formula has been used and change the reference properly.
6) Table 1 is too long and a bit confusing. I suggest dividing the whole content into three different tables: one with clinical and biochemical characteristics of the study cohort at hospital admission; one with comorbidities and one with disease-related features. The last one can be insert also as supplementary material.
7) the authors classify the whole cohort according to the clinical outcome (mild, moderate and severe/death). It would be useful to insert an adjunctive table with clinical characteristic of the study cohort according to different outcomes.
8) it would be of certain importance if the authors could improve the characterisation of the study cohort adding more clinical information, such as blood pressure or BMI, and parameters that better define kidney function, such as BUN or electrolytes, and systemic inflammation as rCP, if available in their dataset
9) Among the therapeutical approaches that have been proposed for patients with Sars Cov2 pneumonia who require hospital admission, antiviral therapy (Remdesivir) is currently a cornerstone and need to be initiated as soon as possible. Reduced kidney function (but also elevated liver enzymes) can limit the access to this treatment. So elevated serum creatinine could negatively impact prognosis also because it exclude patients from an efficacy treatment. The authors should include a comment about this in discussion.
10) As shows in Table 1 patients who die during hospitalisation are significantly older than patients discharged, have much more comorbidities and present higher serum creatinine at hospital admission (even if eGFR is quite preserved for age). Looking at Table 2, is worth noting that patients with baseline creatinine > 1.12 mg/dl (that emerged as cut off point for prediction of in hospital mortality), named “case group”, still present higher age and more comorbidities. Although the authors report that patients in the case group showed higher rate of severe clinical outcomes and in-hospital mortality, no significant differences emerged in PaO2/FiO2 ratio and days from symptom onset at hospital admission, which are two well documented markers of disease severity and worse prognosis. Moreover, in the multivariate analysis, older age, CCI and creatinine, which are somehow interrelated, emerged as the only independent predictors of mortality.
Thus, with available data, we can’t conclude that creatine itself predicts clinical outcome and not just marks older age that is comprehensibly associated with mortality. Authors should stress this concept in discussion.
Author Response
Dear Editor,
We re-submit our paper “PROGNOSTIC VALUE OF CREATININE LEVELS AT ADMISSION ON DISEASE PROGRESSION AND MORTALITY IN PATIENTS WITH COVID-19 - AN OBSERVATIONAL RETROSPECTIVE STUDY” (manuscript n°: pathogens-2474851), modified according to the suggestions of the Reviewers.
ANSWERS TO THE COMMENTS OF THE REVIEWER 1
Point 1: The paper dose not respect the submission guidelines provided by the Journal, in particular: abstract should include a brief background (before stating the aim of the study) and should not include headings; references (for journal articles) should be reported as: Author 1, A.B.; Author 2, C.D. Title of the article. Abbreviated Journal Name Year, Volume, page range. Please check instructions for the authors and correct bibliography
Answer: We thank the reviewer for the suggestion. We modified accordingly the text.
Point 2: Reference number 4 is not about mortality in specific groups of COVID patients but talk about prediction role of liver enzymes at hospital admission on disease progression and clinical outcome. The authors should remove it and cite a proper paper.
Answer: Following the suggestion of the reviewer, we removed the reference.
Point 3: Number 17 and number 22, are duplicated reference, please correct and renumber.
Answer: Following the suggestion of the reviewer, we eliminated the duplicated references.
Point 4: In the “introduction” section, line 73-82 should be removed.
Answer: We modified the introduction as suggested by the reviewer.
Point 5: the authors enrolled 1357 patients admitted to a COVID unit in Campania from March 2020 to May 2021. This long time-period embraces the first three Italian pandemic waves in which the clinical course of the disease and therapeutic approach changed significantly. The authors should possibly specify in the “study design” the setting of care (did they include also Intensive Care Units?) and at least oxygen therapy (conventional oxygen supply or NIMV?) since it is an indirect marker of disease severity.
Answer: According to the suggestions of the reviewer, in the “study design” section of the new manuscript we included the data required.
Point 6: the authors says that eGFR was calculated according to the latest guidelines (line 120). Anyway, the reference proposed (number 11) does not refer to guidelines. Please, specify which formula has been used and change the reference properly.
Answer: We thank you for the suggestion; we are sorry for the mistake. We used for the evaluation of eGFR the 2021 CKD-EPI. We modified the text accordingly, including the reference.
Point 7: Table 1 is too long and a bit confusing. I suggest dividing the whole content into three different tables: one with clinical and biochemical characteristics of the study cohort at hospital admission; one with comorbidities and one with disease-related features. The last one can be insert also as supplementary material.
Answer: We thank you for the suggestions. We modified accordingly the text.
Point 8: the authors classify the whole cohort according to the clinical outcome (mild, moderate and severe/death). It would be useful to insert an adjunctive table with clinical characteristic of the study cohort according to different outcomes.
Answer: Following the suggestion of the reviewer, we added this analysis in Supplementary data (Supplementary table 2).
Point 9: it would be of certain importance if the authors could improve the characterisation of the study cohort adding more clinical information, such as blood pressure or BMI, and parameters that better define kidney function, such as BUN or electrolytes, and systemic inflammation as rCP, if available in their dataset
Answer: We thank you for suggestion. However, the suggested variables were available for a little proportion of patients enrolled.
Point 10: Among the therapeutical approaches that have been proposed for patients with Sars Cov2 pneumonia who require hospital admission, antiviral therapy (Remdesivir) is currently a cornerstone and need to be initiated as soon as possible. Reduced kidney function (but also elevated liver enzymes) can limit the access to this treatment. So elevated serum creatinine could negatively impact prognosis also because it exclude patients from an efficacy treatment. The authors should include a comment about this in discussion.
Answer: As suggested by the reviewer, we evaluated this point bin the discussion section of the new manuscript.
Point 11: As shows in Table 1 patients who die during hospitalisation are significantly older than patients discharged, have much more comorbidities and present higher serum creatinine at hospital admission (even if eGFR is quite preserved for age). Looking at Table 2, is worth noting that patients with baseline creatinine > 1.12 mg/dl (that emerged as cut off point for prediction of in hospital mortality), named “case group”, still present higher age and more comorbidities. Although the authors report that patients in the case group showed higher rate of severe clinical outcomes and in-hospital mortality, no significant differences emerged in PaO2/FiO2 ratio and days from symptom onset at hospital admission, which are two well documented markers of disease severity and worse prognosis. Moreover, in the multivariate analysis, older age, CCI and creatinine, which are somehow interrelated, emerged as the only independent predictors of mortality. Thus, with available data, we can’t conclude that creatine itself predicts clinical outcome and not just marks older age that is comprehensibly associated with mortality. Authors should stress this concept in discussion.
Answer: Following the suggestion of the reviewer, we discussed this point in the new manuscript.
We thank the Reviewers for helping us to improve our paper.
We hope that the paper is now worthy of publication in “Pathogens”
Best regards,
Prof Nicola Coppola
Reviewer 2 Report
Russo and colleagues investigated the association between creatinine levels and the progression of COVID-19 disease and mortality in 1357 admitted adults without underlying chronic kidney disease. The findings of this study reinforce previous research demonstrating that serum creatinine levels can serve as a prognostic indicator for mortality in COVID-19 patients. Moreover, the study provides a specific threshold (1.12 mg/dL) for creatinine levels at admission that can predict mortality outcomes.
Overall, the study is clearly written and presented and uses adequate methodology. However, as the authors objectively state the study presents some important limitations. The main one is the availability of only one creatinine value (at admission) that could have been influenced by a number of factors, including dehydration, concurrent medication etc..., and no additional values during the hospital stay.
In addition to this limitation, there are some major issues of the study that should be addressed:
1) The authors state that no patient was vaccinated for COVID-19 due to the recruitment period that overlapped with the start of the vaccination campaign in Italy, this limitation also has a possible impact on the applicability of the findings to vaccinated patients, which represents the majority of patients nowadays, this should be made clearer.
2) The authors also state that the patients did not receive antiviral or monoclonal treatment due to the recruitment period. However, in 2020-2021, at least some treatments were available in Italy (eg. baricitinib, remdesivir, convalescent plasma) in addition to corticosteroids. Remdesivir, in particular, has been shown to be neutral on renal function (PMID: 34113086, 37128413, 36964312, 36848366) after reports of nephrotoxicity. This should be discussed to offer a more in depth analysis of concurrent factors that may have affected the prognosis and progression of the disease.
3) In Table 2 the authors define an abnormal creatinine value above 0.95 mg/dl. What is the reason for this cut-off? It may be more useful to classify patients according to eGFR rather than creatinine, ideally using the KDIGO stages of CKD.
4) How was the cut-off value of creatinine calculated? Was the Youden index analysis used?
5) The authors cite the work of Chen et al. (REF #11) as the only work that investigated the role of creatinine levels on disease progression. It is true that the work did not identify a cut-off value for creatinine, however, it classified patients as having elevated values if they had a creatinine level above <64.0 μmol/L; elevated creatinine, >104.0 μmol/L (or 1.18 mg/dl), which is in line with the cut-off values found by Russo and colleagues. This may be worth mentioning.
Here are a few minor suggestions for the authors:
1) Lines 73-82 are from the “instruction for authors”, please remove them.
2) To improve the readability of Tables the authors may consider removing the notations in the first column (eg. median (Q1 Q3) or N (%)) and report them in the footnote of the tables.
3) Table 4: correct “N°(%) of patients with abnormal serum creatinine value**” with “Abnormal serum creatinine value” as this refers to a categorical variable and not to the number of patients.
4) Supplementary Figures should have legends.
The quality of English language used in the study is satisfactory.
Author Response
Dear Editor,
Dear Editor,
We re-submit our paper “PROGNOSTIC VALUE OF CREATININE LEVELS AT ADMISSION ON DISEASE PROGRESSION AND MORTALITY IN PATIENTS WITH COVID-19 - AN OBSERVATIONAL RETROSPECTIVE STUDY” (manuscript n°: pathogens-2474851), modified according to the suggestions of the Reviewers.
ANSWERS TO THE COMMENTS OF THE REVIEWER 2
Point 1: The authors state that no patient was vaccinated for COVID-19 due to the recruitment period that overlapped with the start of the vaccination campaign in Italy, this limitation also has a possible impact on the applicability of the findings to vaccinated patients, which represents the majority of patients nowadays, this should be made clearer.
Answer: We thank you for the suggestion. We modified the discussion section accordingly.
Point 2 : The authors also state that the patients did not receive antiviral or monoclonal treatment due to the recruitment period. However, in 2020-2021, at least some treatments were available in Italy (eg. baricitinib, remdesivir, convalescent plasma) in addition to corticosteroids. Remdesivir, in particular, has been shown to be neutral on renal function (PMID: 34113086, 37128413, 36964312, 36848366) after reports of nephrotoxicity. This should be discussed to offer a more in depth analysis of concurrent factors that may have affected the prognosis and progression of the disease.
Answer: As required by the reviewer, we included a paragraph accordingly in the manuscript.
Point 3: In Table 2 the authors define an abnormal creatinine value above 0.95 mg/dl. What is the reason for this cut-off? It may be more useful to classify patients according to eGFR rather than creatinine, ideally using the KDIGO stages of CKD.
Answer: We thank you for the suggestion. We identified this cut-off considering our laboratory range for creatinine levels.
Point 4: How was the cut-off value of creatinine calculated? Was the Youden index analysis used?
Answer : Following the suggestion of the reviewer, we specified in the new manuscript that to identify the cut-off in our population we have used Youden index.
Point 5: The authors cite the work of Chen et al. (REF #11) as the only work that investigated the role of creatinine levels on disease progression. It is true that the work did not identify a cut-off value for creatinine, however, it classified patients as having elevated values if they had a creatinine level above <64.0 μmol/L; elevated creatinine, >104.0 μmol/L (or 1.18 mg/dl), which is in line with the cut-off values found by Russo and colleagues. This may be worth mentioning.
Answer: We are grateful for your comment and modified accordingly the text
Point 6: Lines 73-82 are from the “instruction for authors”, please remove them.
Answer: We thank you for the suggestion. We modified accordingly.
Point 7: To improve the readability of Tables the authors may consider removing the notations in the first column (eg. median (Q1 Q3) or N (%)) and report them in the footnote of the tables.
Answer: We modified the text as suggested by the reviewer.
Point 8: Table 4: correct “N°(%) of patients with abnormal serum creatinine value**” with “Abnormal serum creatinine value” as this refers to a categorical variable and not to the number of patients.
Answer: Thanks for the suggestion. We have modified the text accordingly.
Point 9: Supplementary Figures should have legends.
Answer: We thank you for the comment. Supplementary figure legends were reported in the supplementary data.
We thank the Reviewers for helping us to improve our paper.
We hope that the paper is now worthy of publication in “Pathogens”
Best regards,
Prof Nicola Coppola
Reviewer 3 Report
First of all, we would like to thank the authors for submitting this paper on: an observational study of the use of Creatinine as a prognostic value for mortality at hospital admission in patients with COVID-19.
Although the study is very well written, with an interesting and different methodology, but it is important to recognize that currently, after the worldwide vaccination of COVID-19 with a low incidence of occurrence of this disease, it is not very attractive and interesting to assess the progression and mortality of COVID-19 with a parameter such as creatinine.
Knowing all the biases contained in this serum marker to assess renal function. In this aspect I would recommend more than assessing a cut-off point of serum Cr, it would be more attractive in my opinion to assess changes in the variation of this parameter and thus assess the impact on mortality.
It would also be important to modify some data in the manuscript:
1.- To remove in the introduction section the 4th paragraph (where it is explained how to make an introduction).
2.- In Table 1 please change the position of the results, place the number of patients available in the right row, not the intermediate one, because when reading it, it is a complex result.
3.- In the same table 1 please separate the table by different parameters, for example divide it in basal characteristics, COVID-19 symptomatology, analytical results at hospital admission.
Good quality English, no recommendation from my side.
Author Response
Dear Editor,
We re-submit our paper “PROGNOSTIC VALUE OF CREATININE LEVELS AT ADMISSION ON DISEASE PROGRESSION AND MORTALITY IN PATIENTS WITH COVID-19 - AN OBSERVATIONAL RETROSPECTIVE STUDY” (manuscript n°: pathogens-2474851), modified according to the suggestions of the Reviewers.
ANSWERS TO THE COMMENTS OF THE REVIEWER 3
Point 1: Knowing all the biases contained in this serum marker to assess renal function. In this aspect I would recommend more than assessing a cut-off point of serum Cr, it would be more attractive in my opinion to assess changes in the variation of this parameter and thus assess the impact on mortality.
Answer: We agree with your point of view, but our dataset was created with the aim to highlight the clinical and laboratory data at admission that predict the outcome of COVID-19. We are grateful for your advice and will consider it for future studies.
Point 2: To remove in the introduction section the 4th paragraph (where it is explained how to make an introduction).
Answer: We Following the suggestion of the reviewer, we modified the introduction section of the new manuscript.
Point 3: In Table 1 please change the position of the results, place the number of patients available in the right row, not the intermediate one, because when reading it, it is a complex result.
Answer: We thank you for the suggestion. We modified accordingly in the text.
Point 4: In the same table 1 please separate the table by different parameters, for example divide it in basal characteristics, COVID-19 symptomatology, analytical results at hospital admission.
Answer: We modified the table according to the suggestion of the reviewer.
We thank the Reviewers for helping us to improve our paper.
We hope that the paper is now worthy of publication in “Pathogens”
Best regards,
Prof Nicola Coppola
Round 2
Reviewer 3 Report
Thank the authors for resolving the recommendations requested by the reviewers. For my part the article can be published